# Light Pollution and Oxidative Stress: Effects on Retina and Human Health

**DOI:** 10.3390/antiox13030362

**Published:** 2024-03-18

**Authors:** Rocío Salceda

**Affiliations:** Instituto de Fisiología Celular, Departamento de Neurodesarrollo y Fisiología, Universidad Nacional Autónoma de México, C.P. 04510, Mexico City 70-253, Mexico; rsalceda@ifc.unam.mx

**Keywords:** blue light, oxidative stress, retina, photoreceptor cells, ganglion cells, skin, photo damage, melatonin, LED displays, light pollution

## Abstract

Visible light refers to the frequencies within the electromagnetic spectrum that humans can see, encompassing radiation with wavelengths falling between 380 nm to 760 nm. The energy of a single photon increases with its frequency. In the retina, photoreceptor cells contain light-sensitive pigments that absorb light and convert it into electrical stimuli through a process known as phototransduction. However, since the absorption spectrum of photoreceptors closely aligns with blue light (ranging from 400 to 500 nm), exposure to high light intensities or continuous illumination can result in oxidative stress within these cells, leading to a loss of their functionality. Apart from photoreceptor cells, the retina also houses photosensitive ganglion cells, known as intrinsically photosensitive retinal ganglion cells (ipRGCs). These cells relay information to the suprachiasmatic nucleus in the brain, playing a crucial role in modulating melatonin secretion, which in turn helps in synchronizing the body’s circadian rhythms and responses to seasonal changes. Both, ipRGCs and skin possess a peak sensitivity to blue wavelengths, rendering them particularly susceptible to the effects of excessive blue light exposure. This study delves into the consequences of excessive illumination and/or prolonged exposure to blue light on retinal function and explores its implications for human health.

## 1. Introduction

In recent decades, significant changes in human lifestyles have been driven primarily by industrialization and global modernization. Consequently, various forms of pollution, including air pollution, noise pollution, water pollution, and more recently, light pollution, have emerged as critical sources of concern, impacting climate emergency, ecological issues, and human health [1].

Light pollution, or light at night (LAN), has been steadily increasing in severity. The adoption of more efficient and cost-effective light-emitting diode (LED) technology for street lighting, combined with the growth of the global population and economy, has led to the excessive use of artificial lighting. This has not only contributed to heightened energy consumption and greenhouse gas emissions but has also had a detrimental impact on the natural night sky [1].

LED lighting emits a broad spectrum with a notable blue component [1,2]. While it is important to note that the sun is a primary natural source of blue light, exposure to LED lighting should be viewed as cumulative. This cumulative exposure encompasses both outdoor and indoor spaces, as well as light emitted by electronic displays.

In addition to the social and economic ramifications of light pollution, there are significant potential health risks, particularly to the retina and skin. Light, when absorbed by specific molecules known as chromophores, initiates chemical reactions that involve the transfer of energy (protons or electrons) to other molecules. This renders them chemically reactive and, in turn, capable of inducing oxidative stress.

In animal organisms, evolution has led to the development of specialized light-sensing cells, enabling them to recognize neighboring organisms and respond to stimuli. However, light can also induce significant changes in cell physiology due to its effects on various compounds that can become activated, resulting in the production of highly reactive molecules. When produced in excess, these molecules can lead to oxidative stress, causing substantial damage to cells.

Vision serves as the primary sensory system for humans and many other animals. This complex process commences in the retina of the eye, where photoreceptor cells convert absorbed light energy into electrical stimuli. These stimuli are then transmitted to retinal ganglion cells and further relayed to the brain, where they are processed and translated into images. This intricate process enables organisms to perceive their surroundings and respond to them. Notably, retinal ganglion cells also send axons to brain regions unrelated to visual processing such as the suprachiasmatic and supraoptic nuclei, which regulate the body’s circadian rhythms. ipRGCs also project to the supraoptic nucleus, which participate in fluid homeostasis, social behaviors, and appetite, respectively. Consequently, damage to retinal cells can result in profound alterations in visual perception, potentially leading to blindness, as well as affecting non-visual functions.

Excessive exposure to light can generate high levels of reactive oxygen species (ROS), culminating in oxidative stress and cellular damage. Thus, this review explores various lines of evidence concerning the impact of light pollution on retinal cells and its implications for human physiology.

### 1.1. Light and Oxidative Stress

Light encompasses the wavelength frequencies within the electromagnetic spectrum that are visible to the human eye, ranging from 380 to 760 nm. Light consists of discrete energy packets called photons, each of which is considered a quantum particle with negligible mass. Shorter wavelengths with higher frequencies possess greater energy, and among visible light, blue light (400–500 nm) stands out as the most energetic, constituting 25% of solar radiation [2] and holding the highest energy within the visible light spectrum [3].

While light is essential for life, it can also pose a significant risk to cells and human health. Light photons can be absorbed by atoms and molecules known as chromophores, prompting electrons to transition between energy levels. Photosensitizers, which are light-absorbing molecules, can either donate electrons to adjacent molecules or extract hydrogen atoms from them, resulting in the generation of free radicals and ROS (Figure 1).

Free radicals are atoms or molecules with unpaired electrons, rendering them highly reactive, as exemplified by hydrogen peroxide (H_2_O_2_) and hydroxyl radicals (•OH). ROS, on the other hand, are byproducts of regular oxygen metabolism, such as singlet oxygen (1O_2_) and superoxide radicals (O•^−^) [4,5]. Excessive ROS production can lead to photodamage, ultimately culminating in cellular damage or even cell death [6,7,8]. It is worth noting that low levels of ROS play roles in cellular signaling and homeostasis. Depending upon the source of ROS and cell type, ROS signaling participates in metabolic regulation, cell growth, and autophagy by inducing AMP-activated protein kinase activity. However, an overabundance of ROS results in oxidative stress, causing the oxidation of various cellular components, impairing their normal functioning [9,10]. To counter the effect of excessive ROS, cell possess numerous molecules call antioxidants, which neutralize ROS by donating an electron to the unpaired oxygen molecule (Figure 2).

Blue light can be absorbed by a range of biomolecules, including nicotinamide adenine dinucleotide phosphate (NADP, NADPH), porphyrins, DNA, proteins, and lipids [11]. Consequently, excessive light exposure can generate unwarranted ROS, inducing oxidative stress and cellular injury [8,11,12,13].

Due to global industrialization, artificial lighting has become an integral part of daily life. One significant concern is the transition from incandescent lamps to more cost-effective and efficient light-emitting diodes (LEDs). LEDs are characterized by a prominent blue light component with relatively high energy and absorption depth [12,13]. Furthermore, the widespread adoption of LED-backlit liquid crystal displays (LCDs) in various electronic devices such as computers and smartphones has contributed to issues related to light pollution, excessive artificial lighting, increased energy consumption, and greenhouse gas emissions [14,15,16]. In fact, electronic devices emit approximately 30% of their radiation as blue light, while indoor lighting sources emit between 6% and 40% of their radiation in the blue light spectrum [17]. Consequently, in addition to concerns regarding air, noise, and water pollution, it is important to recognize that light pollution has adverse effects on the environment, biodiversity, human health, and overall quality of life [14,15,16,17,18].

### 1.2. Effects of Light on Retina Photoreceptors

Among the light-triggered reactions, retinal phototransduction is arguably the most pertinent to human vision. The retina, a neural tissue, possesses the ability to detect light and generate visual signals for the brain. This metabolically active tissue comprises five distinct types of neuronal cells, with photoreceptor cells being paramount. These photoreceptors have the capacity to absorb light and transform it into an electrical signal through a process known as phototransduction. The signal is subsequently relayed via bipolar cells to ganglion cells, which further transmit signals to the brain for image processing. Horizontal and amacrine cells facilitate lateral communication among retinal neurons [19].

Additionally, the retina houses three types of glial cells: Müller glia cells, astrocytes, and microglia [20]. Furthermore, retinal vascular endothelial cells (VECs) are responsible for supplying blood to the retina. The retinal pigment epithelium (RPE), in close proximity to photoreceptor cells, serves as part of the retinal blood barrier and offers structural and functional support to the neural retina [21].

The retina features two kinds of photoreceptor cells, cones and rods, responsible for vision in bright and low-light conditions, respectively. Cones can be further categorized into three types (L, M, S) based on their ability to absorb specific wavelengths (long, medium, and short) [22], wavelengths (long/red, medium/green, and short/blue) [22]. The photoreceptors have specialized outer segments consisting of membranous discs containing the protein rhodopsin, which is bound to a chromophore, 11-cis retinal. In the absence of light, the photoreceptor membrane potential remains at approximately −40 mV, primarily due to the action of Na+ and Ca2+ ion channels, creating a dark current [23]. When exposed to light, 11-cis retinal is converted into all-trans retinal, which then dissociates from the opsin and is transported to the RPE. There, enzymatic reactions regenerate 11-cis-retinal [24,25,26,27] (Figure 3).

Simultaneously, the activated opsin, metarhodopsin II, interacts with the G protein transducin, and its α subunit activates a phosphodiesterase enzyme. This enzyme, in turn, hydrolyzes cGMP into 5’GMP [28,29]. The reduction in cGMP levels results in the closure of the dark current and subsequently leads to the hyperpolarization of the photoreceptor cell [29] (Figure 3).

Approximately 11,000–15,000 photons are absorbed per rod per second [30], a count that can increase in the human retina based on one’s workplace and activities in well-lit environments, including medical imaging of the retina [31,32]. However, excessive and/or prolonged exposure to light can induce damage to the retina by causing photochemical harm [8,33,34]. Pioneering studies have shown that light intensities two to three times higher than those of normal illumination can result in damage to visual cells in nocturnal animals [35,36,37]. Moreover, a plethora of studies have demonstrated that retinal cells are particularly vulnerable to short-wavelength light, notably blue light [38,39,40], with photoreceptors and the retinal pigment epithelium being among the most affected [37,41,42,43,44,45].

Retinal lesions are characterized by the degeneration of photoreceptor outer segments, accompanied by the generation of ROS, ultimately leading to cell death [46,47]. Furthermore, exposure to sunlight has been linked to the development and/or worsening of age-related macular degeneration [48,49].

Despite certain eye structures, such as the cornea, aqueous humor, lens, and vitreous humor, absorbing some light wavelengths, more than 65% of blue light at 460 nm is transmitted to the retinas of children under 9 years of age, rendering them especially susceptible to retinal damage [50]. Therefore, the retina is highly prone to light-induced damage due to its continuous exposure to light, the concentration of chromophores in photoreceptor cells [24,26], high levels of the polyunsaturated lipid docosahexaenoic acid (DHA) [37,51,52], and heightened mitochondrial activity. Notably, photoreceptor outer segment discs contain high concentrations of unsaturated fatty acids, including the highest level of DHA in the body, rendering these cells particularly susceptible to oxidative stress [51,52]. Remarkably, all-trans-retinal is considered a photosensitizer, and its accumulation can lead to photochemical damage [53,54]. The bisretinoid A2E, a byproduct formed from phosphatidyl ethanolamine and an excess of all-trans-retinal, is the primary component of lipofuscin, a product of photo-bleached rhodopsin that accumulates with age [55,56,57,58]. Furthermore, lipofuscin is particularly sensitive to blue light absorption, and when exposed to light, it generates ROS [59]. In this regard, studies involving rats exposed to full-spectrum white LEDs and blue LEDs (at 460 nm) resulted in free radical production and apoptotic and necrotic photoreceptors, indicating photochemical injury [31,54,60,61,62]. Moreover, high levels of A2E have been associated with macular degeneration [40]. Additionally, it has been demonstrated that all-trans-retinal leads to the activation of NADPH oxidase, which in turn can increase oxidative stress, trigger RPE apoptosis, and induce an inflammatory response [54]. Furthermore, the photoreversal bleaching of rhodopsin can enhance its photon absorption capacity [32,54], thereby increasing oxidative stress.

Under normal physiological conditions, the mitochondrial respiratory chain is responsible for approximately 90% of ROS generation within a cell [63]. Photoreceptor cells and RPE cells contain abundant mitochondria housing various chromophores that have the potential to trigger photodamage [64,65]. Among these chromophores, flavin, and porphyrin possess absorption peaks within the blue light range. The transfer of electrons from these photosensitizers to oxygen can generate ROS, specifically singlet oxygen. Subsequent reactions may yield superoxide radicals, hydrogen peroxide, and hydroxyl radicals [11,66]. Additionally, these chromophores can initiate the oxidation of various molecules (e.g., carbohydrates), ultimately leading to H_2_O_2_ production, which contributes to lipid and protein oxidation [67,68,69].

In this context, long-term exposure to blue light has been shown to significantly elevate ROS levels and the expression of Bax (bcl-2-like protein 4) and trigger the mitochondrial apoptosis pathway [70,71]. High-intensity blue LED light exposure also induces photoreceptor apoptosis, necrosis, and retinal gliosis [72,73].

Remarkably, Nrf2 (nuclear factor erythroid 2-related factor 2), the master regulator of redox state, exhibits high expression levels in the retina [74] and is known to protect against oxidative stress and inflammation [75]. Notably, sulforaphane has a protective effect under blue light exposure through the activation of this transcription factor [75]. In addition to Nrf2, the retina boasts high concentrations of non-enzymatic and enzymatic antioxidants, as well as the pigments lutein and zeaxanthin, which serve as ROS scavengers and guard against light-induced damage [58,76].

Furthermore, aside from ROS production, prolonged exposure to blue light leads to an increase in the Unfolded Protein Response (UPR), as evidenced by the phosphorylation of PERK (Protein kinase R-like endoplasmic reticulum kinase) and the upregulation of ATF4 (Activating Transcription Factor), resulting in the increased expression of apoptosis-related genes [77]. Long-term blue light exposure has also been observed to induce endoplasmic reticulum stress, DNA, and inflammatory damage in both photoreceptor cells and RPE [54,78,79,80].

### 1.3. Impact of Light on Retinal Pigment Epithelium

Derived from the neural tube that gives rise to the neural retina, the RPE is essential in providing oxygen and nutrients to the retina, and in regulating ion and fluid balance. It is actively involved in the daily phagocytosis of shed photoreceptor outer segments, recycling of visual pigments, and secreting various growth factors necessary for the construction and upkeep of the choroid and photoreceptors [21,81,82].

Like the retina, the RPE is abundant in mitochondria and harbors light-absorbing molecules such as melanin, lipofuscin, and retinoids, which make it prone to photochemical damage [31]. Nonetheless, similar to the retina, the RPE has a significant antioxidant capability that protects against A2E oxidation and ROS production [58,79].

The primary mechanism of damage induced by all-trans-retinal involves the generation of singlet oxygen. Additionally, the accumulation of all-trans-retinal can lead to the formation of Schiff base adducts with phosphatidylethanolamine. In culture RPE cells, all-trans-retinal accumulation triggers the activation of NADPH oxidase and TLR-3 (Toll-like receptor 3). This activation contributes to increased oxidative stress, apoptosis, and inflammatory responses [63]. Furthermore, treating RPE cells with all-trans-retinal results in heightened ROS levels within the mitochondria and endoplasmic reticulum, as well as elevated mRNA expression of Nrf2 and its target proteins [83].

Age-related macular degeneration (AMD), the foremost cause of legal blindness in the developed world, is associated with the RPE’s accumulation of phototoxic substances, notably lipofuscin. This accumulation promotes oxidative stress, leading to the degeneration of cones [84]. Although the contribution of lipofuscin and its photosensitizing chromophores to AMD development remains a topic of discussion, the accumulation of A2E in post-mitotic RPE cells as a result of aging and excessive illumination poses a risk factor for cellular damage [85].

Studies on human primary RPE cell cultures exposed to visible light for 3 h have shown a mild reduction in mitochondrial respiratory activity and mitochondrial DNA damage. Consistently, isolated RPE mitochondria subjected to light exposure produced singlet oxygen, superoxide anion, and the hydroxyl radicals [86].

Research utilizing the human cell line ARPE-19 highlighted the phototoxic effects of blue light. Exposure of ARPE-19 cells to blue light led to increased ROS production and caspase 3/7 activity, which correlates with decreased cell viability [87]. In contrast, exposure to red or green light did not cause damage to these cells. However, in vivo experiments are needed to substantiate these findings.

These effects were intensified in ARPE-19 cells preloaded with A2E, underscoring the molecule’s role as photosensitizer. Additionally, an increase in both A2E and vascular endothelial growth factor (VEGF) production was observed in cultured ARPE-19 cells following light exposure [88].

Exposure of ARPE-19 cells to blue LED light resulted in significant ultrastructural changes in mitochondria, mitochondria depolarization, increased ROS production, and a loss of cell viability. This was accompanied by enhanced formation of advanced glycation end products (AGEs) and delay in the cell cycle [84]. Blue light exposure also led to the upregulation of phospho-p38-MAPK (mitogen-activated protein kinase), a protein known to be activated under cell stress and apoptosis. Notably, these effects were absent when cells were exposed to red light.

Remarkably, exposure of both albino and pigmented mice to 1 or 3 days of blue LED-light caused damage to the RPE and photoreceptors. In contrast, exposing pigmented mice to three days of fluorescent light did not result in retinal damage [89]. Similar outcomes were observed in both albino and pigmented rats exposed to domestic luminance levels using various LEDs, compact fluorescent lamps, and fluorescent tubes. This study indicated that white LEDs but not compact fluorescent lamps, caused retinal degeneration, and breakdown of the RPE barrier, as evidenced by albumin leakage. Interesting these changes were not observed with green LED exposure. Consequently, these findings strongly suggest that the white-LED may pose a risk of retinal toxicity at domestic and occupational luminance levels, potentially contributing to the pathogenesis of AMD [45].

### 1.4. Photosensitive Ganglion Cells and Physiological Rhythms

Organisms rely on light perception for object recognition and guidance in their actions, serving as a mechanism to sense ambient light intensity. This sensing of light is crucial for synchronizing the circadian clock with the natural solar day and influencing seasonal physiological rhythms. While traditional photoreceptor cells in the retina are fundamental for visual processing, a novel class of retinal ganglion cells that contain the photosensitive pigment melanopsin (Opn4) has been identified [90,91]. These intrinsically photosensitive retinal ganglion cells (ipRGCs) are widespread across the animal kingdom, with six distinct subtypes (M1-M6) identified in mice. In the human retina, ipRGCs account for approximately 1% of the total ganglion cell population [92,93].

Similar to rods and cones, ipRGCs convert light into electrical signals. Melanopsin binds to 11-cis retinal and exhibits peak sensitivity at a wavelength of 460 nm [94]. Upon light activation, melanopsin interacts with a G protein, triggering a phospholipase (PLC β4), which then initiates the production of second messengers IP3 (inositol triphosphate) and DAG (diacylglycerol). These messengers facilitate the opening of nonselective cation channels (TRP), resulting in cell depolarization [95,96,97]. The phototransduction mechanism is maintained through the recycling of the chromophore retinal in RPE and Müller glial cells.

Importantly, melanopsin protein levels increase at the onset of light exposure, peaking towards the end of the day and diminishing in darkness [98,99]. The decline in melanopsin levels allows these cells to respond more sensitively to changes in light intensity rather than contrast, enabling them to maintain continuous activity.

Intrinsically photosensitive retinal ganglion cells (ipRGCs) primarily project to the suprachiasmatic nucleus (SCN), the central regulator of the environmental light/dark cycle through its control of melatonin secretion from the pineal gland. Melatonin, a key neuromodulatory hormone, peaks during darkness and plays an essential role in aligning behavioral and physiological activities with environmental light cues [100,101].

Given melanopsin’s peak absorption at 460 nm, exposure to blue light during the day is pivotal in regulating various physiological functions. Light therapy, leveraging this principle, is effectively used to treat conditions such as seasonal affective disorder and circadian sleep disorders [102].

Besides the SCN, ipRGCs extend their axons to several brain areas, including the dorsal lateral geniculate nucleus (dLGN), the olivary pretectal nucleus (OPN), and the supraoptic nucleus. These areas are integral to the pupillary light reflex, as well as influencing social behaviors and appetite, respectively [103,104,105,106,107,108,109,110]. Thus, light’s influence extends beyond visual perception, impacting pupillary responses, eating habits, energy metabolism, alertness, sleep patterns, mood, and cognitive functions [110,111,112,113,114,115,116,117,118,119,120,121,122]. Disruptions in these neural circuits due to inappropriate light exposure can heighten the risk of developing a range of disorders, including cardiovascular diseases, obesity, and mental health conditions [118,119].

Epidemiological studies have indicated that chronodisruption is linked with an increased incidence of several diseases, including diabetes, cognitive and affective impairments, and some types of cancer [11,12,118,119]. Furthermore, night-time workers who are exposed to light at night face a higher risk of cardiovascular disease, hypertension, obesity, depression, and cancer [118,119,120,121]. Additionally, evidence suggest that LED displays can suppress melatonin at night and disrupt biological rhythms [116,117,118,119,120,121,122]. These disruptions are thought to be caused by changes in ipRGCs signaling, which affects the SCN. The SCN regulates various biological rhythms and through adrenocorticotropin, influences the secretion of corticosteroids involved in metabolism, immune response, cardiovascular function, and reproduction [13,14,15,122,123,124,125].

Exposure to blue light-emitting LEDs has been shown to decrease melanopsin expression and damage ipRGCs [126]. LED exposure in animal models has led to mitochondrial damage, reduced dendritic arborization of ipRGCs, increased retinal GFAP immunoreactivity, and apoptosis in the outer nuclear retinal layer [127].

In terms of peripheral effects, melatonin is reported to play a role in blood pressure regulation [16,128]. Evidence suggests an increase in melatonin release during mania and a decrease during depression [14,124]. Some studies have indicated that mood episodes in bipolar disorder are triggered by disturbances in circadian rhythms. Genetic studies have also identified a connection between bipolar disorder and specific circadian genes such as CLOCK and TIMELESS [14,15,124,125]. Consequently, darkness has been suggested as a mood stabilizer for manic patients. Recent research has investigated melatonin agonists as pharmacological interventions to stabilize circadian rhythm disruptions in these patients. Additionally, blue light-blocking treatments, such as the use of amber-tinted glasses, have been found to improve sleep in manic patients [17,129]. Interestingly, bright light therapy has shown positive effects on mood and physiological parameters, suggesting the need for further research to understand these processes fully [18,130]. Moreover, the use of mobile phones in bed before sleeping has been found to negatively affect mood, sleep quality, memory, and concentration [19,130,131]. These findings underscore the importance of maintaining adequate light exposure to preserve synchronized rhythms and promote health and wellness.

Remarkably, mice exposed to dim blue light for four consecutive weeks (12:12 h, dark/dim blue light) exhibited oxidative stress, microglia activation, increased mRNA expression of inflammatory proteins, and a decrease in hippocampal neuron numbers. These effects were linked to impaired spatial learning and memory [20,131].

### 1.5. Effects of Light on the Skin and Uvea

The skin and eyes are the only human organs directly exposed to light radiation. The skin serves as both a barrier and a defense mechanism against the external environment. Blue light exposure in the skin triggers mediators of skin aging, DNA damage, and apoptosis in both normal and cancerous cells [22,132].

In addition to the previously discussed effects of light, sunlight is well-recognized as a human carcinogen. Both natural and artificial light directly affect the skin, with blue light, in particular, having the capacity to penetrate more deeply (0.07 to 1 mm) than ultraviolet rays (UV) [126,127,133,134]. Notably, the epidermis expresses melanopsins OPN2, OPN3, and OPN4, which are activated by blue light. This activation triggers TRP channels, leading to the activation of calcium/calmodulin-dependent protein kinase II (CAMKII), and results in changes in gene transcription [128,135].

Moreover, apart from UV-induced damage, blue light is associated with accelerated skin aging, increased hyperpigmentation of skin melanocytes, and the development of dark spots [126,127,128,129,133,134,135,136,137].

In addition to blue light photodamage, lifelong exposure to sunlight plays a significant role in the development of skin aging characteristics, though further in vivo studies are required to fully understand the mechanisms and action spectrum of photoaging in humans [43,135]. In keratinocytes, blue light-activated opsin leads to a decrease in per1 transcription, suggesting that skin cells can regulate clock gene production in response to light exposure. At night, the focus shifts towards repairing daytime damage [45,46,136,137]. Thus, it is suggested that blue light could disrupt the skin cells’ nocturnal rhythm, essential for cell regeneration and repair. Human skin is impacted not only by ultraviolet radiation but also by the blue light wavelengths emitted by sunlight, electronic devices, and light-emitting diodes [44,138].

Furthermore, human keratinocytes exposed to high-irradiance blue light demonstrated an increase in ROS production, changes in mitochondrial morphology, and a significant decrease in the expression of clock genes compared to exposure to red light or under low-irradiance conditions. Blue light exposure also led to cell death and cell cycle arrest at the G2/M phase [28,139].

Additionally, in non-tumorigenic human keratinocytes, blue light exposure induced oxidative stress, which in turn triggered endoplasmic reticulum stress and promoted apoptosis [32,140]. Similarly, rats exposed to blue light (8000 lx) for 8 h experienced skin damage and degeneration mediated by oxidative stress, endoplasmic reticulum stress, and apoptosis [33,141]. Therefore, due to its capacity to increase DNA damage, blue light can cause erythema, hyperpigmentation, and photodamage [34,35,138,142] and has been utilized successfully to eliminate tumor cells [36,37,38,143,144,145].

However, the effect of blue light on cancer induction has also been studied. Skin cancer was induced in hairless mice exposed to blue LED light daily (10 min/day) for one year. The irradiation with blue light increased the expression of proliferation markers Ki-67 and cyclin D1. Additionally, blue light exposure led to the production of ROS, inflammatory proteins, and increased migration of neutrophils and macrophages involved in carcinogenesis in the skin [22,132]. Notably, skin cancer was not induced in mice exposed to green or red LED light.

Uveal melanoma is the most common primary intraocular malignancy in adults, originating from melanocytes within the uvea. Ultraviolet exposure, combined with specific skin pigment gene polymorphisms, is a significant factor in its development [23,82,146,147]. However, when cultured human uveal melanoma cells were exposed to blue light, there was a notable increase in their proliferation rate, an effect that was mitigated by using a blue-light-filtering lens [25,148]. Additionally, the development of ocular tumors was observed in pigmented Long Evans rats following blue light exposure [26,149]. Similarly, human uveal melanoma cell lines (92.1, MKT-BR, OCM-1, SP6.5) subjected to blue light exhibited a significant increase in cell proliferation rate. Remarkably, the application of blue light filtering intraocular lenses neutralized this effect [27,140].

Moreover, melanoma cells exposed to blue LED light for 1 and 2 h per day experienced increased ROS production and DNA damage; cells irradiated with blue light displayed depolarized mitochondrial membranes, elevated caspase-3 activity, and an increase in melanin synthesis. Blue light also suppressed cell proliferation and induced cell cycle arrest [29,30,150,151].

Despite these effects, the anti-proliferative properties of blue light may be leveraged for treating certain dermatological conditions, including acne, psoriasis, and precancerous lesions [39,152]. Blue light therapy has shown promising results in acne management [40,153], and positive outcomes have been noted in treatments with topical retinoids for mild skin conditions, such as psoriasis [130,131,132,153,154,155,156,157,158,159,160,161,162].

Many cosmetic products contain retinoid-based compounds, such as retinyl palmitate (RP), aimed at protecting the skin or promoting skin responses that repair damage caused by sunlight. It is important to recognize that retinoids can increase light phototoxicity through photochemical reactions [131,158]. Naturally occurring retinoids can affect gene expression, acting as both antioxidants and pro-oxidants through the photochemical production of reactive oxygen species (ROS). Additionally, they can serve as agonists or antagonists for retinoic acid or retinoid-X receptors, which are involved in regulating gene transcription impacting embryonic development, tissue homeostasis, metabolism, and cellular differentiation [132,159]. Dysregulation of these receptors has been associated with a variety of conditions, including cancer, autoimmune diseases, metabolic disorders, and inflammatory diseases [132,159,160,161]. Moreover, blue LED light has been observed to significantly enhance hair growth in patients with androgenetic alopecia [41,162]. Similarly, blue LED devices have shown positive effects on wound healing and pain reduction in patients with chronic wounds [42,163].

These varying outcomes underscore the necessity for additional research to explore the photodecomposition products of retinoids and their potential phototoxic effects on cells.

## 2. Discussion

Concerns regarding the safety of LED light sources have escalated with the widespread use of cellphones, tablets, laptops, and desktop computers, especially during the COVID-19 pandemic [164]. Notably, in 2020, the World Health Organization officially recognized digital technology addiction as a global issue [119]. This addiction is marked by compulsive, habitual, and uncontrollable use of digital devices and excessive engagement in online activities, leading to significant sleep deprivation, emotional distress, and memory impairments a problem that has grown during the COVID-19 pandemic [119,130,154,160,164,165,166].

Despite this information, there is currently insufficient evidence to conclusively determine the direct effects of LED light sources when used normally at domestic intensity levels or as backlights in screen devices. Consequently, more epidemiological, clinical, and fundamental research is required to assess their risk factors in AMD, aging, and overall health, especially for vulnerable groups like children and adolescents [167].

In the interim, it is crucial to minimize excessive light exposure. Several measures have been proposed to achieve this [13,168,169,170]. Exposure to morning sunlight is recommended to help maintain synchronized rhythms, along with the use of blue light cutting glasses and blue light cutting base makeup. Blue-blocking glasses, also known as amber glasses, reduce the light component that activates ipRGCs, thereby aiding in the onset of dim-light melatonin. Their effectiveness in improving sleep and treating bipolar disorder, major depression, and postpartum depression has been documented [124,129].

Furthermore, to alleviate digital eye strain from prolonged use of electronic devices, recommendations include ergonomic practices, adequate ambient lighting, limiting daily screen time to about 4 h, taking regular breaks, and using blue-light filtering glasses with an anti-reflective coating.

Additionally, a diet rich in antioxidants, including fresh fruits and vegetables high in vitamin C, vitamin E, carotene, lutein, and zeaxanthin, is strongly recommended [10], as these nutrients offer photoprotective benefits [118,125].

## 3. Conclusions

While light plays a crucial role in visual function, it is important to recognize that exposure to intense natural or artificial light, particularly in the blue light spectrum, can be detrimental to retinal photoreceptor cells due to the induction of oxidative stress. Additionally, excessive illumination can impact intrinsically photosensitive retinal ganglion cells (ipRGCs), which transmit signals to the brain to regulate melatonin secretion. Consequently, this disruption of biological rhythms can lead to various adverse effects on human health and overall quality of life. Moreover, excessive illumination promotes skin aging and might favor cancer development.

Hence, it is of paramount importance to identify these negative consequences and develop strategies aimed at mitigating the effects of light pollution on human health. Furthermore, public education on this issue and the promotion of reduced unnecessary illumination are essential steps in addressing this concern.

### Future Research Directions

Considerable evidence suggests that excessive illumination, particularly in the form of blue light, can result in retinal cell injury, leading to significant changes in visual perception, circadian rhythms, and even blindness. This raises significant concerns regarding the excessive use of indoor lighting and electronic displays. However, the precise mechanisms by which blue light-induced reactive oxygen species (ROS) production leads to retinal cell death remain unclear. Therefore, further studies are required to elucidate the photodamage caused by different LED emitters and the effects of acute and chronic light exposure.

As LEDs are poised to become the primary light source in our domestic environments, it is imperative to gain a comprehensive understanding of the mechanisms underlying the effects of light on cells to ensure necessary protective measures.

The consequences of excessive natural and artificial illumination have been underestimated by modern society. It is crucial to educate the public about these effects to reduce the duration of light exposure and maintain a normal dark-light cycle. Additionally, the establishment of regulations to protect the environment and human health is essential. Exploring more suitable light sources with fewer detrimental effects on biological systems would also be a pertinent avenue of research.

## Figures and Tables

**Figure 1 antioxidants-13-00362-f001:**
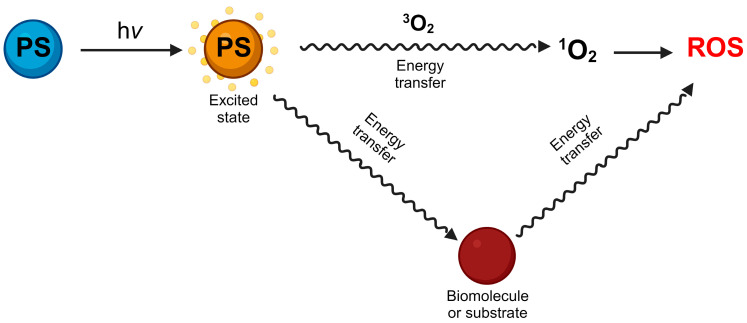
Photosensitizers (PS) absorbs light energy reaching higher excited states, this energy is transferred to oxygen or other molecules giving rise to ROS.

**Figure 2 antioxidants-13-00362-f002:**
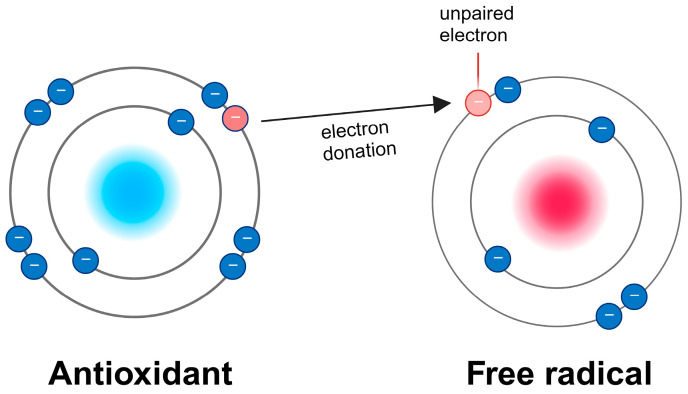
Antioxidants protect from oxidative stress by donating an electron to the unpaired oxygen.

**Figure 3 antioxidants-13-00362-f003:**
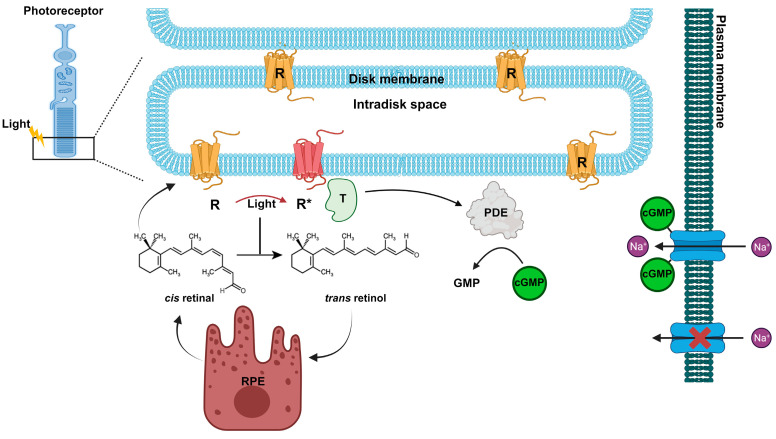
Rhodopsin (R) is concentrated in the disc membranes of rod photoreceptor cells. Rhodopsin contains the chromophore 11-cis retinal, which under light stimulation is converted in all-trans retinal. The photoactivated Rhodopsin (R*) triggers a signal transduction cascade, leading to activation of the phosphodiesterase (PDE) and a consequent decrease in cGMP and closure of the Na^+^ dark current. While retinoid pathway follows the renewal of 11-cis retinal at the retinal pigment epithelium (RPE).

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
