# Peer review of "Light Pollution and Oxidative Stress: Effects on Retina and Human Health"

_antioxidants, 2024, doi:10.3390/antiox13030362_

Round 1

Reviewer 1 Report

Comments and Suggestions for Authors

The paper titled "Light Pollution and Oxidative Stress: Effects on Retina and Human Health" presents an interesting review of the impact of blue light and light pollution on retinal function, vision, and human health. However, I have a few suggestions and remarks.

In the introduction section, it is advisable to specify which non-visual functions are affected by damage to retinal cells caused by blue light and the underlying mechanisms involved.

Additionally, it would be beneficial to explore existing studies on the influence of blue light on human and animal behaviour, psychological changes, and its consequences on biological rhythms. Including these research findings in the manuscript and providing a more in-depth analysis of their results and conclusions would enhance the comprehensiveness of the review. The manuscript should also encompass data on the potential influence of visible light wavelengths other than blue light on the retina, visual function, and overall health.

Furthermore, it is important to address whether prolonged exposure to blue light has any correlation with the development of eye melanoma. Additionally, clarifying potential differences in the impact on uveal melanoma and conjunctival melanoma due to their distinct exposure to light is recommended.

To offer more practical insights, the manuscript should include more detailed recommendations for mitigating the impact of blue light. Providing suggestions on ways to protect the eyes and maintain optimal visual functions would be valuable to readers interested in practical applications of the research.

Author Response

The paper titled "Light Pollution and Oxidative Stress: Effects on Retina and Human Health" presents an interesting review of the impact of blue light and light pollution on retinal function, vision, and human health. However, I have a few suggestions and remarks.

In the introduction section, it is advisable to specify which non-visual functions are affected by damage to retinal cells caused by blue light and the underlying mechanisms involved.

Non visual functions mediated by ipRGCs, which can be affected by blue light are mention in the introduction.

Additionally, it would be beneficial to explore existing studies on the influence of blue light on human and animal behaviour, psychological changes, and its consequences on biological rhythms. Including these research findings in the manuscript and providing a more in-depth analysis of their results and conclusions would enhance the comprehensiveness of the review. The manuscript should also encompass data on the potential influence of visible light wavelengths other than blue light on the retina, visual function, and overall health.

Thanks for signalling that.  In the revised manuscript, those points are extended and information related to circadian rhythms, behaviour and mood is shown.  The effect of other wavelengths (red and green) was also indicated.

Furthermore, it is important to address whether prolonged exposure to blue light has any correlation with the development of eye melanoma. Additionally, clarifying potential differences in the impact on uveal melanoma and conjunctival melanoma due to their distinct exposure to light is recommended.

Thanks for your comment, it is a very interesting point. Although a relative few information exists on that, which may appera in controversy, Recent studies on this aspect are mentioned in a new caption,       

To offer more practical insights, the manuscript should include more detailed recommendations for mitigating the impact of blue light. Providing suggestions on ways to protect the eyes and maintain optimal visual functions would be valuable to readers interested in practical applications of the research.

Thanks for mention that. This point is mentioned at the discussion, and some indications for eye protection are shown in the revised manuscript.

Reviewer 2 Report

Comments and Suggestions for Authors

The manuscript entitled "Light pollution and oxidative stress: effects on retina and hu- man health", submitted for publication as a review article in Antioxidantes, addresses an interesting and very topical subject, the effect of excessive exposure to light on the retina and skin.

Although well written, the manuscript has some aspects that need to be corrected and others altered/added. Thus,

1) The authors state in the abstract that light refers to wavelengths between 380 and 760 nm. These wavelengths refer only to visible light and not to all light, so they should correct the information.

2) Still on wavelengths, the authors refer in line 9 to the 380-760nm range and in line 71 to the 400-760nm range. The authors should standardise the range.

3) The authors should review the entire manuscript and correct all the subscripts and superscripts used (H2O2, Ca2+, 1O2, etc....).

4) The legend of figure 3 does not correspond exactly to the information contained in the figure, but is more complete. I suggest that the authors complete the figure with the relevant information in the caption.

5) The authors describe (line 134) the types of cones and state that "Cones can be further categorised into three types based on their ability to absorb specific wavelengths (red, green, and blue)", implying that cones are classified as red, green and blue. This classification has fallen into disuse and today we use the S, M and L designations for short, medium and long wavelengths. The authors should rephrase the sentence so that this interpretation cannot occur.

6) In line 156 the authors state that "The human retina typically absorbs approximately 1012 to 1015 photons each day". This figure is not correct (is it per second?) and is not supported by the references presented (30,31).

7) The discussion isn't really a discussion. In fact, it's not often that a review article has a discussion section. The authors should redo this part, because in the discussion and conclusions the authors repeat a lot of the information presented previously, without adding much new.

8) The authors should add a section on the effects of blue light on RPE cells. There are plenty of studies showing the damaging effect of blue light on these cells and the contribution this could have on the development of AMD.

9) In the title, the authors state that the article deals with the effects of light pollution on the retina and human health. Apart from the retina, the authors only present the effects on the skin. I believe that the authors should rework the title, adapting it better to the content of the manuscript.

10) The authors only present harmful effects on the skin. There are studies that show that, depending on the dose and wavelength, blue light can be beneficial and even used to treat acne (e.g. PMID 32979787).

Author Response

The manuscript entitled "Light pollution and oxidative stress: effects on retina and hu- man health", submitted for publication as a review article in Antioxidantes, addresses an interesting and very topical subject, the effect of excessive exposure to light on the retina and skin.

Although well written, the manuscript has some aspects that need to be corrected and others altered/added. Thus,

1) The authors state in the abstract that light refers to wavelengths between 380 and 760 nm. These wavelengths refer only to visible light and not to all light, so they should correct the information.

Thanks for your precise comment. I corrected that point along the text.

2) Still on wavelengths, the authors refer in line 9 to the 380-760nm range and in line 71 to the 400-760nm range. The authors should standardize the range.

Thank you for your thorough review of the manuscript. I did the adequate standardization as you suggested.

3) The authors should review the entire manuscript and correct all the subscripts and superscripts used (H2O2, Ca2+, 1O2, etc....).

Thanks for your observation, these mistakes were corrected along the entire manuscript.

4) The legend of figure 3 does not correspond exactly to the information contained in the figure, but is more complete. I suggest that the authors complete the figure with the relevant information in the caption.

Thanks for your suggestion. The figure was modify according to the legend, and the information is stated in the legend´s figure.

5) The authors describe (line 134) the types of cones and state that "Cones can be further categorised into three types based on their ability to absorb specific wavelengths (red, green, and blue)", implying that cones are classified as red, green and blue. This classification has fallen into disuse and today we use the S, M and L designations for short, medium and long wavelengths. The authors should rephrase the sentence so that this interpretation cannot occur.

You are quite right, sorry for the confusion. It was corrected to amend it.

6) In line 156 the authors state that "The human retina typically absorbs approximately 1012 to 1015 photons each day". This figure is not correct (is it per second?) and is not supported by the references presented (30,31).

Thanks for the accurate remark. Indeed, this information was wrong. It was amended and the adequate reference is shown.

7) The discussion isn't really a discussion. In fact, it's not often that a review article has a discussion section. The authors should redo this part, because in the discussion and conclusions the authors repeat a lot of the information presented previously, without adding much new.

Thank you for your valuable proposal. An Introduction section is request in the author´s instruction. But in agreement to your comment the discussion was reduced.

 8) The authors should add a section on the effects of blue light on RPE cells. There are plenty of studies showing the damaging effect of blue light on these cells and the contribution this could have on the development of AMD.

Following your comment, a particular section referring the effect of blue light on RPE was included.

9) In the title, the authors state that the article deals with the effects of light pollution on the retina and human health. Apart from the retina, the authors only present the effects on the skin. I believe that the authors should rework the title, adapting it better to the content of the manuscript.

Thanks for the feedback. In this review information of the light effects on eye and skin is shown, because they are the only structures directly in contact with light. However, because blue light affect not only conventional photoreceptors but also ipRGCs, which input directly to the the hypothalums, particularly controlling rhythms, light´s effects on these cells can lead to alterations in a variety of functions; therefore may affecting general health.  Some data supporting this idea is extended in the revised manuscript (Photosensitive Ganglion Cells and Physiological Rhythms section).

10) The authors only present harmful effects on the skin. There are studies that show that, depending on the dose and wavelength, blue light can be beneficial and even used to treat acne (e.g. PMID 32979787).

You are right, thanks for the remark. There is evidence that blue light can be used for treating certain dermatological conditions and even precancerous lesions. That is information is expanded in the revised manuscript (Effects of light on the skin and uvea section), following the Reviewers request.

Reviewer 3 Report

Comments and Suggestions for Authors

This review paper by dr. Salceda recapitulates the events elicited by light exposure (mainly blue light) in the retina and the damaging consequences of excessive exposure on the different photoreceptive cells of the retina, namely rod and cone photoreceptors and the intrinsically photosensitive retinal ganglion cells. The data reviewed here suggest the need of an increased attention towards the effects of light pollution on the human health.

I would suggest some changes that may improve the manuscript and, maybe, increase the interest for this paper.

- In discussing the relationships between light exposure and oxidative stress, some mention is made of the physiologic effects of ROS and of the fact that the cell possesses a nyumber of antioxidant molecules, however I don’t think this is sufficient. In my opinion, a better description, in dedicated chapters, shoud be made of (i) the physiologic role played by ROS in the cells, (ii) the mechanisms of ROS-induced damage to cell components when ROS levels become excessive, (iii) the cellular antioxidant systems. With this information, the impact of light on the cell redox system and the pathologic consequences should be well understood by the reader.

- There are a couple of mentions of the relationships between retinal light exposure and AMD, but the issue is not discussed. I think that a chapter could be dedicated to the effects of light exposure in retinas that are already suffering for other pathologies, such as AMD, diabetic retinopathy, retinitis pigmentosa, etc.

- The additiojn of these new chapter could be compensated by the deletion of the final chapters (“Effects of light on the skin”, which is out of context, and Discussion, Conclusions and Future directions, which do not provide significant material for the paper)

- Please revise all the abbreviations. Some of the are defined multiple times, some others are definedbut they are never used, or the same abbreviation is used for different names. Although not substantial, this defect results quite annoying to the reader.

Other comments:

Line 117: The chapter also considers RPE

Lines 132-155: the existence of rods and cones and the mechanism of phototransduction is available in any physiology textbook and it is not necessary to repeat it here.

Fig. 3: The sentence “The photoactivated Rhodopsin triggers …” does not belong to the figure.

Line 165-167: It is not clear whether light induces degeneration of outer segments, and therefore ROS generation, which leads to cell death, or light induces ROS generation, which, in turn, causes outer segment degeneration and, ultimately, cell death.

Line 167-168: the mention of AMD is out of context. The relationships between light exposure and retinal disease should be treated in a separate section.

Lilne 174, 175: It is not clear why the high DHA levels should facilitate retinal light damage, and the role of the “heightened mitochondrial activity” should be better explained. I see that these subjects are reconsidered later (lines 189-201): I suggest to concentrate the discussion on DHA or mitochondria in a single paragraph.

Line 183-184: I suggest that the relationships of light exposure with AMD and, possibly, with other retinal diseases are examined separately.

Line 202, “In this context”: Do you mean that the reported effect of light on retinal cell apoptosis is likely to be mediated by light-induced oxidative stress? If so, it should be stated more clearly.

Line 209: I don’t think that “oxidative stress transcription factor” is an appropriate definition of Nrf2.

Line 228: it should be [81] and not [82]. On the other hand, ref 81 does not seem to fit with the sentence on lines 222-224, since it does not mention A2E. Please revise the references related to these points.

Lines 250-252: It is not clear how the lower levels of melanopsin (by the way, lower than … what?) should be related to light intensity more than to contrast sensitivity.

Lines 270-274: This paragraph generates some confusion. The “context” is that of the effects of blue light on ipRGCs. Ref 112 reports reduced melanopsin and reduced dendritic arborization size and complexity in ipRGCs (and not just in RGCs). Instead, mitochondrial damage has been observed in RGCs (and not specifically only in ipRGCs), while increased GFAP and apoptosis in the ONL have nothing to do with ipRGCs.

Lines 285-310: the effects of light on the skin are out of context. There is no link with the reported damaging effects of light in the retina and a review of the effects of light exposure on skin pathologies would require a separate manuscript. This chapter should be deleted.

Lines 313-355: this discussion is just a summary of the previous parts and it does not provide a real discussion. Lines 357-374 may serve as concluding remarks.

Lines 377-408: The conlcusion paragraph and the future research directions do not add any new consideration and are not necessary.

Author Response

Please find the responses in the attachment. Thank you.

Round 2

Reviewer 1 Report

In this updated manuscript, the changes that have been made are visible. The author has responded to the suggestions and enhanced the quality of the manuscript. Therefore, the manuscript can be accepted in its current form.

In this updated manuscript, the changes that have been made are visible. The author has responded to the suggestions and enhanced the quality of the manuscript. Therefore, the manuscript can be accepted in its current form.

Reviewer 2 Report

The revised version of the manuscript includes corrections to the points made in the comments and many of the suggestions made. I therefore consider that this new and improved version fulfills the necessary conditions to be accepted for publication.

Nothing to add.

Reviewer 3 Report

Some of the suggested changes were not applied, but the Author explained and justified her choice, which I respect.

I have no further comments.